# Analyzing Ad Exposure and Content in Child-Oriented Videos on YouTube

## ABSTRACT

As a popular choice for video and entertainment streaming, YouTube hosts a large audience, including children, who form a growing proportion of its users. Despite separate "made for kids" labelling and stricter moderation of these videos, inappropriate advertising remains a concern as it threatens the safety of YouTube for young viewers. This paper is the first comparative measurement study that explores how advertisement exposure and content vary across child-oriented videos on YouTube. We do this by conducting a cross-regional advertisement analysis on highly viewed "made for kids" labelled content across a total of five strictly regulated and five unregulated countries. A second front of comparison is carried out between ad patterns on unlabelled and labelled child-oriented videos. Our analysis reveals that the safety of a child's YouTube experience is shaped significantly by their external environment and surrounding child safety policies. There also exists a gap in YouTube ad and child protection policy enforcement, indicated by the presence of unlabelled child-oriented content with weak ad regulation. We discuss the implications of inappropriate exposure on children and suggest policy and implementation measures to mitigate this threat.

## 1 INTRODUCTION

Hosting over 2.6 billion unique users, the YouTube platform is now the predominant choice for video and entertainment streaming, with a total of 1 billion hours watched per day [14]. A growing proportion of these viewers are children (aged under 13). According to a 2020 Pew Research survey involving US parents, 81% of 4,591 participants say their child watches YouTube regularly [41]. As such, YouTube also hosts an overwhelming abundance of child-oriented content, with *Cocomelon*, primarily known for its animated nursery rhymes, as the second most subscribed channel on the platform, boasting over 150 million subscribers [50].

With this level of child exposure, there is growing concern about the prevalence of age-inappropriate ads, especially on videos for younger viewers, leaving impressionable children susceptible to negative influences on their behaviour and development [34]. Despite YouTube maintaining that the platform is intended for users over 13, citing its more regulated counterpart, YouTube Kids, for younger audiences – the rise of the use of shared devices, family content, and the fact that a majority of the videos on the platform can be viewed without signing in, reflects the need for ensuring child safety on the main YouTube application, which hosts a significantly larger pool of users [22]. According to a study released in December 2020 by nScreenMedia and WildBrain Spark, a survey of 3,000 parents reveals that while two-thirds of children watch YouTube Kids, 70% of kids use YouTube, establishing the heightened usage of the main app for children [24].

Child protection on YouTube also remains a litigated domain, with YouTube paying USD 170 million in fines for violation of the US children privacy law, COPPA, for collecting child user data for targeted advertising [27]. In its aftermath, in November 2019, the platform claimed strong policy measures to make the platform safer for children, with the enforcement of a "made for kids" label on child-oriented videos restricting regular features such as comments, and notifications, and enabling greater ad regulation [18].

In this paper, we conduct a measurement study to understand how ad exposure and content varies across child-oriented videos on YouTube. In doing so, we *quantify* how much ad content children are exposed to on YouTube and analyse what proportion of this content is irrelevant or age inappropriate, guided by YouTube's own publicly stated policies. Our comparison is conducted across regions with a strong policy presence in child online safety, and those without any regulatory protection. A second front of comparison is across child-oriented content that is labelled as "made for kids" on the platform, against content that is oriented for children but remains unlabelled, with subsequently higher risk to weaker ad protection.

We carry out cross regional analysis on a set of 750 top viewed "made for kids" videos, across 5 high policy regions (US, UK, France, Germany, Sweden), and 5 low policy environments (Bangladesh, Sri Lanka, Pakistan, Morocco, Venezuela). The choice of these countries was informed by DQ Institute's Child Online Safety Index for 2022 and evaluating the regulations in place for child advertising in each country [25]. High policy countries have been regarded as having safer online environments for children and stricter child advertising policies. Ad metadata is collected on each video across all the ten regions to enable a comprehensive cross country analysis. For comparison, we construct a second set of 750 popular videos with *unlabelled* child-oriented content on YouTube, each of which is viewed from the same five high policy and five low policy countries. Ad information is gathered, and thereby compared to the ad patterns in the equivalent labelled dataset collected from the same regions. Our resulting dataset, which can be found here, comprises 1,500 unique videos (i.e. 750 labelled "made for kids" videos and 750 unlabelled videos) – each viewed from ten different countries, summing to 15,000 videos analyzed – and a total of 24,148 ads.

Table 1 summarizes the key insights from our study. In particular, we find that despite a strong policy presence, highly regulated environments tend to show more ads per video in comparison to low policy regions. However, the total ad duration per video is 15.0% shorter in high policy regions compared to low policy regions. The high policy areas better emulate the YouTube Kids safety experience, with 73.5% of ads meeting the YouTube Kids limits for ad durations [18]. We find that 49.5% and 77.2% of unique unskippable ads, in high policy and low policy regions respectively, do not comply with YouTube's publicly stated limits (15 seconds), going up to 1.8 hours in length. Our study shows that external policy presence drastically affects nature of ads on "made for kids" content. High policy regions have 3.6% inappropriate ads, and 19.2% child-oriented ads whereas a stark 28.8% of ads in low policy regions have inappropriate content, and only 8.3% are child-oriented. Majority of ads are irrelevant,

| Key Insights | Description |
|---|---|
| (1) Role of policy presence in exposure to inappropriate ads | In high policy regions, child-oriented videos, labelled by YouTube as "made for kids", had 3.6% inappropriate ads whereas low policy regions had 28.8% inappropriate ads. This suggests that external policies influence exposure to inappropriate ads. |
| (2) Widespread deviation from YouTube ad specifications | 49.5% and 77.2% of the *unique* ads in high policy and low policy regions, respectively, do not comply with YouTube's publicly stated limit (15 secs), going up to 1.8 hrs in length. |
| (3) Ad duration across low and high policy regions | In less regulated areas, ads tend to be longer, with a 15% increase in average ad watch time compared to high policy regions (184 secs in low policy regions vs. 159 secs in high policy regions). |
| (4) Need for more ubiquitous labelling of child-oriented content | Our dataset includes 750 child-oriented videos (with a cumulative view count of 9.98 billion) that have not been labelled as "made for kids" by the platform. Out of these, 60.4% targeted non-English speaking audiences, revealing a significant gap in the current "made for kids" labelling practice between English and other languages. |

**Table 1: Key insights from our measurement study.**

limiting return for advertisers who do not wish to target children, and increasing unnecessary exposure for young users.

Finally, our unlabelled dataset consists of a representative sample of 750 videos meant for children, 60.4% of which is intended for non-English speaking audiences, with 33.6% of all videos native to the subcontinent, indicating a gap in the current labelling regimen.

These insights have important implications for platform providers and advertisers. For platform providers, tighter regulation and enforcement of ad policies is crucial to minimize the risk of exposure to age-inappropriate ads. Some of the ways this can be done is by limiting ad categories like "Music" on child-oriented content, enforcing limits on ad frequency and duration relative to the main video length, and ensuring more relevance of provisioned ads for young age audiences. More proactive solicitation of feedback on ads from users (including co-viewing parents)—possibly within-player—can also help in mitigating inappropriate exposure. Additionally, YouTube can benefit from replicating several YouTube Kids safeguards on the main app, which include the use of ad bumpers to distinguish ad and main video content, as well as disabling outbound ad links or showing ad click prompts to alert users about leaving YouTube. There is also a need for ubiquitous labelling, marking all child-oriented videos as "made for kids," through standardized labelling methods that rely less on self-reporting. This may require more manual review, assessing video interactions like comments as well as broader channel evaluations. These measures can help ensure platform-led safety for its growing child user base.

In addition, greater responsibility needs to be placed on advertisers to provide ad metadata, such as accurate "made for kids" labelling, and relevant descriptions and titles, to enable greater standardised and scalable assessment of ad content, and ensure self-reported regulation that promotes safe and age-appropriate experiences on YouTube. Opportunity also exists for advertisers that target children to collaborate with the platform and provide appropriate ad content that can reach a large child viewership while improving the effectiveness of advertising efforts.

Overall, our contributions can be summarised as follows:

(1) We present the first large-scale analysis of YouTube ads on child-oriented content, across ten countries to determine the role policy presence surrounding child online safety plays in ad exposure. We have shared our dataset, it can be found here.

(2) We define a comprehensive coding scheme to classify the age appropriateness and relevance of video ads qualitatively, providing a total of 3,000 tagged video ads for use in future work for inappropriate ad detection.

(3) We build two more robust datasets: the first comprising 750 popular unlabelled videos oriented for child viewership, with a cumulative view count of over 9.98 billion; and the second

consisting of over 24,000 total ads from across ten different countries. These help us in measuring the impact of regulation on ad patterns, and can be used by key stakeholders to assess child safety on YouTube.

(4) We provide policy and implementation recommendations for YouTube as a platform provider to ensure a safer and more homogeneous experience for all child users.

## 2 METHODOLOGY

We now describe the methodology we employed to (i) identify and collect a comprehensive dataset of labelled and unlabelled child-oriented content on YouTube, (ii) scrape and collect a video ad dataset comprising skippable and non-skippable video ads, in-feed video ads, sidebar ads and overlay ads, and (iii) annotate the video ad dataset to identify ads as (a) inappropriate, (b) child directed and (c) irrelevant for young audiences.

### 2.1 Video Dataset Construction

**Labelled Video Set.** Our methodology for gathering "made for kids" videos on YouTube involves identifying the most popular children's videos worldwide. Noticeably, the Youtube API [31] does not provide access to a ranking of the most-viewed "made for kids" videos. As a result, we utilise data from Social Blade [13], a YouTube-certified user-analytics platform, which maintains a list of the most popular channels with the "made for kids" tag, ranked based on total lifetime views. We select the top 10 most popular videos from the 75 highest-viewed kids channels on this list, by querying the YouTube API [31]. This forms our labelled video dataset, comprising 750 videos, capturing a broad spectrum of content and styles which are likely to attract a large number of young viewers worldwide. The resulting dataset surpasses 360 billion views in total.

**Unlabelled Video Set.** While focusing on "made for kids" channels is a useful starting point for analysing ad patterns on kids' videos, it is also important to consider the wider landscape of child-oriented content on the platform, much of which remains unlabelled. Exploring this unlabelled content provides a comprehensive understanding of the types of videos that current labelling algorithms fail to capture, and the implications for ads featured on this content.

To build a representative dataset for child-oriented unlabelled videos, we use seed search words reflecting popular child interests, some of which include "toys", "kids cartoon", and "barbie", taken from the most popular child-oriented YouTube searches indicated by Google Trends [39]. The results are then manually parsed to find popular channels with unlabelled content, with a minimum threshold of 400,000 views. It is noteworthy that 80.6% of the resulting dataset have above a million views, indicating their popularity.

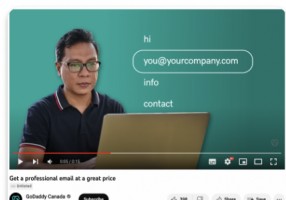 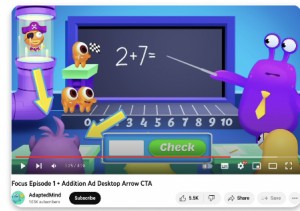 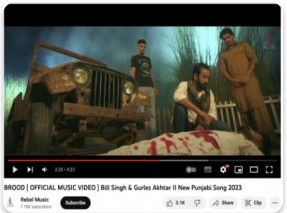 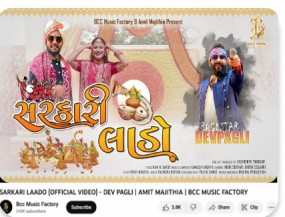

**Irrelevant**  **Child Directed**  **Inappropriate**  **Ambiguous**

**Figure 1: Examples of tagged videos: "irrelevant" (software), "child-directed" (educational content), "inappropriate" (interpersonal violence), and "ambiguous" (language barrier).**

After identifying candidate videos for our unlabelled dataset, two coders assess each video to *confirm* its child-oriented nature. This assessment borrows from YouTube's publicly stated guidelines for determining if a video is "made for kids", where they evaluate video content to ensure one of two standards are met: "(i) children are the primary audience, or (ii) the video contains characters, songs, games, stories, or other subject matter reflecting an intent to target children" [32]. We employ the same mechanism by checking for the presence of common themes in kids content, such as child-oriented animations, in each of the videos we find (Table 8). Additionally, metadata like tags, title, and description are manually analyzed for kids-directed keywords. User comments referencing age are also considered to determine the demographic of viewers.

Simultaneously, we confirm the absence of unsuitable themes from the video content, even if it is visually child-oriented. These include mature elements like violence and sexual innuendos. Any presence of these established negative themes warrants the exclusion of the video from our dataset, even if multiple child-oriented elements are present. The full rubric used for unlabelled video analysis can be found in the Appendix (Table 8). Overall, this approach allows us to carefully construct an unlabelled child-oriented dataset, comprising 750 videos with over 9.98 billion total views, enabling analysis of ads across a breadth of kid-oriented content types on YouTube.

## 2.2 Region Selection

For our study, we select ten countries with varying regulations on child advertising and online safety. First, we study DQ Institute's Child Online Safety Index 2022, which evaluates 100 countries on six fronts of child cyber-safety [25]. Using this index, we identify both high and low-ranked countries, and then analyze their official policies on child advertising to confirm their relative standing [11].

Following this, we finalise five countries with weak regulation for child internet safety (Bangladesh, Pakistan, Sri Lanka, Morocco and Venezuela), and five strong policy environments (US, UK, France, Germany and Sweden). The same video dataset is used across all selected regions for ad collection, ensuring a controlled evaluation of ad patterns independent of varying video choices.

## 2.3 Ad Metadata Collection

Next, we scrape ad data across all videos for further analysis, covering all major ad formats on the platform including (i) skippable and (ii) unskippable video ads, (iii) sidebar ads, (iv) in-feed ads, and (v) banner ads [40].

To automate this process, we utilize a Selenium Webdriver script [7] to play each video. To eliminate the potential influence of user-level information on ad choice, the script launches YouTube in a new logged-out Chrome window, with no previous history, cookies, or user data [4]. We then scrape each ad's unique YouTube-assigned video ID, and any embedded external link as the video plays. Next, we use YouTube Data API [8] to obtain additional metadata like video title, duration, "made for kids" label, and tags for each video ad. This raw data forms the basis of both quantitative and qualitative comparisons in our analysis.

## 2.4 Manual Annotation of Video Ads

To examine content patterns in our ad video dataset, we employ a standard deductive coding scheme [35] that utilises predefined tags to classify the content's age-appropriateness for child audiences. For our study, we consider children as individuals under the age of nine.[1] This matches the age rating for our labelled dataset, as categorized by YouTube Kids, which marks each included channel as intended for children up to 8 years old.

After surveying prior literature, as well as existing commercial tools, we find that there is no existing technology or model available to classify inappropriate audiovisual content for the 0-8 age group. Previous research has often relied on manual tagging processes to address concerns about inappropriate exposure to children [17, 21, 30, 36, 42, 48]. We also study existing YouTube guidelines for unsuitable advertising on "made for kids" content and find that it references broad themes including "Adult and Sexually Suggestive Content", "Scary Imagery", "Profanity and Sexual Innuendo", calling for some degree of human assessment [18].

**Codebook Generation.** Since the larger aim of our work is to measure platform-level enforcement of a safe ad experience for children, we develop a codebook entirely from YouTube's own stipulated policies [18]. This codebook is meant for use by coders to identify inappropriate ad content based on the platform's standards, instead of an independent judgement of the ad's nature. It also borrows from work done by Common Sense Media, a leading independent source for media age ratings. This allows for a more detailed and comprehensive categorization of YouTube's stated broad ad themes, ensuring coverage of all ad types [21]. The complete codebook is available in the Appendix.

Under this qualitative coding scheme, each video ad will have one primary tag from the following list, with examples shown in

---

[1]This approach helps address the challenges of assessing appropriateness within a broader age range like 0-13, as defined by YouTube.

Fig. 1. This is dependent on which tag most dominantly applies, based on a visual and auditory analysis of the content:

- **_Child directed_**: Content containing features intended for individuals below the age of 9.
- **_Irrelevant_**: Content targeting viewers aged 9 or above, with general audience elements irrelevant to younger child interests.
- **_Inappropriate_**: Content with features that deem it unsuitable for viewership by children, which may include but are not limited to suggestive content, violence, scariness, weapons or drugs.[2]
- **_Ambiguous_**: Content that cannot be understood by the coder for accurate labelling, to avoid misinterpretation of information. This may be due to a language difference, or unclear audio.

For each primary tag, we maintain a list of secondary tags that further classify the ad content theme and ensure more fine-grained and accurate labelling. These secondary tags are developed from Common Sense Media's work, while the "inappropriate" category tags are derived from YouTube's publicly stated "restricted" and "prohibited" ad categories for children [18, 21]. For example, the "inappropriate" primary tag may have secondary tags such as physical violence, suggestive content, offensive language, or extreme stunts. The complete coding scheme can be found in the Appendix.

**Test Coding.** In order to ensure the reliability of the manual annotation process, a focus group was conducted with 20 ad videos, from high and low-policy regions, to expand the codebook template and ensure its completeness. Through consensus coding [23], we collectively watched and analyzed these ads, allowing for the emergence of new secondary tags. This also allowed us to develop each secondary tag with descriptive examples, and assign applicable videos for reference. This hybrid approach to qualitative coding has been adopted across past literature to ensure greater rigour [28]. Finally, all authors independently tagged another set of 20 ads, in order to enhance their understanding of the final codebook and to ensure consensus in the tagging process. A target agreement rate of 0.85 was set, which was successfully met.

**Final Coding Process.** Once the codebook was established, a random sample of 150 ad videos was chosen from each of the ten regions for both labelled and unlabelled datasets. This sampling is weighted by the frequency of an ads' appearance.[3] The resulting tagging dataset comprises 3,000 videos. Each video is tagged by two independent coders, with one primary coder, chosen at random to prevent bias. All coders are third and fourth-year university students, aged 21 to 22, and fluent in English, Hindi, Urdu, and Punjabi. Coders watched each ad with audio, identified dominant themes, assigned relevant primary and secondary tags, and added descriptive notes. An ad received a primary tag only if an applicable secondary tag from the codebook matched its content. To establish inter-rater reliability, we use Cohen's kappa [19], maintaining a kappa value of at least 0.85 for each tagging sprint, comprising 150 ad videos from a specific country. The final value was found to be 0.98 across all tags.[4]

## 3 DATA ANALYSIS

In this section, we discuss our findings, both quantitative and qualitative, following an analysis of 24,148 total and 6,355 unique video ads, across 10 regions and 2 content labels. We choose to focus on video format ads, as they constitute 86.8% of our ad dataset, and subsequently elicit the highest level of exposure to children.

We first examine ad patterns on "made for kids" labeled videos across different regions, assessing ad duration and frequency. Then, we evaluate compliance with YouTube's ad limits and measure policy enforcement. The appropriateness and relevance of ads is also analyzed, identifying markers of inappropriate content across regions. We extend this analysis to compare labeled and unlabelled child-oriented videos, considering both strong and weak regulatory environments. This helps measure potential risks for children viewing unlabeled content and explores the impact of labeling on ad provision on YouTube.

### 3.1 Comparison across regions

Below, we compare ad patterns on labelled "made for kids" videos, across high and low policy geographies.

**Ad frequency.** When comparing the breadth of ad exposure, we find that policy presence does not seem to have a strong influence on ad frequency. As such, highly regulated environments tend to show more ads per video in comparison to low policy regions—1.1 ads per video on average in low policy areas increase to 1.8 in high policy, a statistically significant difference (Welch t-test: $p < 0.001$). Low policy regions have 21.4% more ad-free videos compared to high policy areas, where 7.4% of all videos feature 5 or more ads.

This is explained further by a comparison of the ad pool in both regions, with our high policy region dataset having 76.9% more unique ads in comparison to low policy regions. One may attribute this to a difference in digital advertising spend, which on average, is reported to be USD 68.9 billion for high policy regions, and only USD 296 million for low policy regions [46]. In more developed advertising markets, the extent of ad exposure is seen to rise significantly, even in the context of young age audiences and regardless of the strong policy presence. This heightened exposure has the potential to influence child attitudes towards consumption, and can blur the distinction between entertainment and advertising, influencing children's understanding of media content [49].

**Ad duration.** Next, we look at the influence of policy presence on ad duration, defining "ad time" as the cumulative sum of all unskipped ad lengths on a single video. On average, ad time is 15.0% longer in low policy regions compared to high policy (184 secs vs. 159 secs) on the same video dataset ($p = 0.06$). However, low-policy regions have a median ad time of 0 secs versus 32 secs in higher-policy regions, explained by the lower frequency of ads (Mann Whitney U Test: $p < 0.01$). Notably, outliers significantly inflate ad duration values in less regulated regions.

In investigating country-specific ad patterns, Fig. 3 also shows that the distribution of ad time has greater homogeneity across high policy regions. The variance of means increases 4.7× from

---

[2]All secondary tags are explained in the Appendix, based on YouTube guidelines.
[3]The sample size is limited by the minimum number of ads observed for a specific country (169 unique ads appeared on the unlabelled dataset when viewed from Venezuela).
[4]This value indicates "Almost Perfect" level of agreement among raters [38].

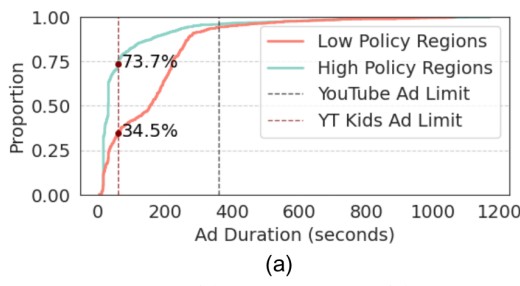
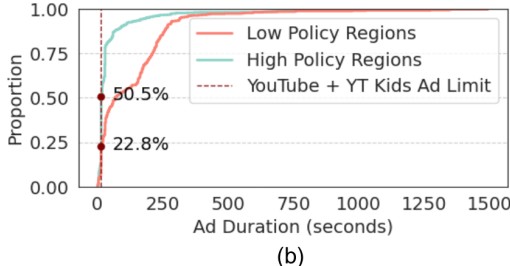

(a)                                                                (b)

**Figure 2: CDFs of (a) skippable and (b) unskippable ad durations across high and low policy regions.**

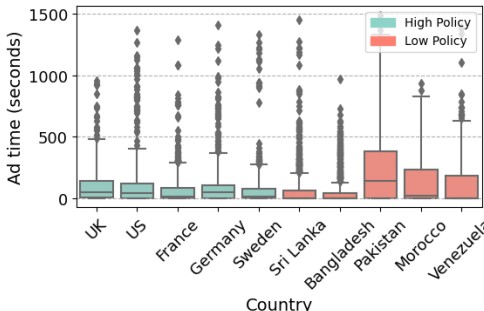

**Figure 3: Ad time per video across high/low policy regions.**

high to low policy regions, indicating more consistent ad provision in higher policy regions, with less extreme values, ensuring more uniform ad exposure on "made for kids" content.

All in all, with dense outliers and varying ad exposure, Fig. 3 indicates the need for more concerted efforts to curb lengthy advertising on child-oriented content. Pakistan is notably anomalous in this sample, with a mean ad time of 488 secs, and an outlier total ad duration of 1.5 hrs, 42× the main video length. Such prolonged exposure risks an over-commercialization of the child viewership experience—a consequence seen to heighten materialism, and potential parent-child conflict [15].

**YouTube Ad policies.** To analyze the influence of the external environment on YouTube's policy enforcement, we examine platform-led ad regulation in different regions. Specifically, we analyze the proportion of unique skippable ads adhering to YouTube's defined time limits: 60 secs for YouTube Kids (YTK) app and 6 mins for the main app [9]. Despite our measurement being based on the main app, compliance with the YTK limit provides a valuable benchmark for evaluating ad exposure against a standard deemed safe by YouTube in child contexts, especially considering that these ads are displayed on *labelled content* also available on YouTube Kids.

Fig. 2a confirms that current advertising practices deviate from YouTube Kids' safety measures, despite the presence of the "made for kids" label. Given the main app's large child audience [24], this poses a significant risk of unnecessary ad exposure, by YouTube's own standards. Regions with stronger child online safety regulations exhibit better adherence to YouTube Kids' safety standards, as evident in Fig. 2a. In these areas, a substantial portion (73.7%) of the skippable ad pool is within YouTube Kids' 60-second limit. Conversely, in low policy regions, *only* 34.5% of ads meet this requirement. The median duration differs notably across geographies: 31 secs in high policy and 160 secs in low policy regions ($p < 0.001$,

Mann Whitney U test). While conformity to the main app's 6-minute limit is prevalent, outliers with ad durations up to 4.1 hrs emphasize the need for heightened platform-wide attention to this issue.

In contrast, examining unique unskippable ads and their adherence to YouTube's 15-second limit [9] in Fig. 2b reveals non-compliance across the board. This deviation appears much earlier in low policy unskippable ads, where only 22.8% adhere to the limit, compared to 50.5% in high policy regions— highlighting the role that external regulation may play in YouTube's ad choices, particularly those that violate policy. The difference in mean ad length, 86.9 secs in high policy vs. 171 secs in low policy ($p < 0.001$), also indicates more controlled ad provisioning in higher policy areas.

This widespread non-compliance has led many users to express frustration on social media [1]. News reports linked this to a YouTube experiment, said to end by September 2022 [37]. However, our dataset, collected from February to May 2023, shows the problem persists even after the experiment's stated end date.

**Ad themes.** Next, we conduct a thematic analysis of 1,500 labelled ads, across both high and low policy regions, maintaining independent coding and strong inter-rater reliability ($\kappa = 0.98$).

*Ad themes based on primary tags.* Fig. 4 confirms that in both regions, "irrelevant" ads are the most prevalent, standing at 76.3% of the annotated ads from high policy regions and 54.9% in low policy regions. This raises questions about YouTube's ad provision and pricing policies, which state that content creators only receive payment if viewers completely watch or engage with the ad [10]. As such, content creators may face financial challenges if the majority of ads are irrelevant, impacting their ability to generate income from their content. Advertisers may also benefit if YouTube reassesses its ad distribution to ensure ads reach their intended audience.

Child-directed ads are more common in high policy countries (19.2%) compared to low policy countries (8.3%). This implies that in regions with stricter regulations, ads are better suited to children's interests, making their viewing experience more relevant. However,

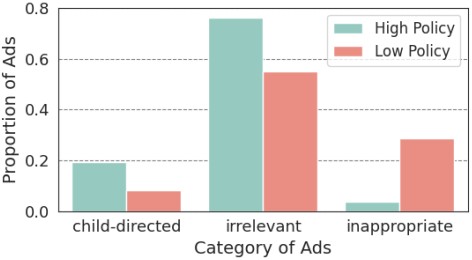

**Figure 4: Ad themes in labelled content.**

| High Policy | | Low Policy | |
|---|---|---|---|
| *Secondary Tag* | *Perc.* | *Secondary Tag* | *Perc.* |
| Scariness | 25.9% | Suggestive Content | 37.2% |
| Suggestive Content | 22.2% | Drinking, drugs, smoking | 14.0% |
| Drinking, drugs, smoking | 14.8% | Display of Weapons | 10.2% |

**Table 2: Common themes in inappropriate ads category.**

child-directed ads still constitute less than one fifth of the tagged ad pool, indicating the need for greater platform-wide efforts for more age-appropriate advertising.

Most concerning is the proportion of "inappropriate" ads across both regions—only 3.6% in high policy regions in comparison to 28.8% of the ads in low policy regions. This suggests that regions with stricter policies are more successful in reducing inappropriate ads for child viewers. The 8× increase in such ads for low policy regions raises major concerns about the safety of child viewers, with almost one in three sampled ads containing inappropriate content, potentially impacting child development and behavior [34]. It also invokes questions about the platform's responsibility to offer a uniformly safe experience for all users, with such gaps pointing to potential prioritization of some audiences' experiences over others.

***Ad themes based on secondary tags.*** Analyzing common themes for inappropriate ads in Table 2 reveals notable differences in the types of content that children are exposed to across different regions. Children in low policy regions are at a higher risk of exposure to "suggestive content" which is a restricted YouTube ad category, defined as "age-sensitive or sexual media content suitable for adult audiences" [18]. This category comprises 37.2% of inappropriate ads, significantly higher than the 22.2% observed in high policy regions. This disparity suggests that in areas with weaker regulation, there may be more permissive attitudes towards certain content types, potentially exposing children to inappropriate material. In contrast, in high policy regions, the predominant theme among inappropriate ads is "scariness", which may evoke fear or distress among young viewers, impacting their emotional well-being [51]. Taken together, this composition emphasizes the need for more comprehensive regulations, accounting for varying inappropriate themes across regions, to ensure a responsible advertising environment.

***Ad categories across regions.*** Considerable disparities also exist in YouTube's own content categorization of ad videos across high and low policy environments. In low policy regions, the top category is "Music" (36.8% of all ads), which often incorporates adult themes, possibly contributing to the higher prevalence of inappropriate ads. This is further supported by the results from qualitative coding, where 53.8% of music ads have been tagged as "inappropriate". In contrast, a large proportion of ads (15.1%) in the high policy dataset are categorized as "Science and Technology" videos. All of the tagged videos that belong to this category have been marked as "irrelevant". This highlights how content type influences the video's appropriateness for child viewers, suggesting the potential to use categories for better ad content moderation.

## 3.2 Comparison across label type

**Unlabelled Dataset Composition.** As per policy, YouTube requires all creators to label their child-directed videos as "made for kids" on uploading. Beyond primary reliance on self-identification,

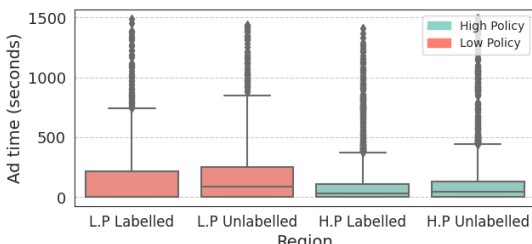

**Figure 5: Ad time per video across content labels in low/high policy regions.**

the platform also claims to employ machine learning to find videos targeted towards children, considering features like kids characters, toys, or games [54]. Using our unlabelled video dataset as a representative sample, with over 9.8 billion views, we aim to highlight potential gaps in this current labeling practice, exploring the types of content that may be commonly missed by the algorithms cited.

Thematically, all videos in the dataset appear to match the nature of the videos required to be labelled as "made for kids" [53]. 46.5% of its videos are kids-oriented moral stories or fairy tales, like "Goldilocks", or "Cinderella". Other prevalent themes include clips or movie trailers from popular kids entertainment, like "Boss Baby", along with videos featuring play sessions with toys like Barbies. Interestingly, we find that 60.4% of the dataset comprises videos for non-English speaking audiences. Specifically, 33.6% of the videos are in languages indigenous to the subcontinent (Urdu, Tamil, Hindi, Bengali, among others), with 9.3% in European languages (including Spanish, Greek, Polish, Hungarian). Our dataset also captures videos with no verbal dialogue, accounting for 115 of the 750 videos.

Understanding the implications of children watching these videos is essential, because their content aligns with labelled videos, and a large proportion comes from regions with consistently high YouTube viewership [45]. We measure this implication in the context of video ads ahead. To do this, we explore ad exposure and content patterns across labelled and unlabelled child-oriented content, both in the context of high and low policy regions.

**Ad Frequency.** We find that there is a 23.9% overall increase in the number of ads per video from labelled to unlabelled content, indicating the label's role in limiting ad exposure to children for enhanced safety on the platform. However, region-specific trends show that ad frequency is more heavily influenced by the external environment. Irrespective of the assigned age label, videos in high policy regions consistently show more ads compared to low policy regions. Even unlabelled videos from low policy regions have a lower average ad count (1.7 ads per video) than labelled videos in strongly regulated areas (1.8 ads per video). This shows that external factors strongly influence ad selection, highlighting the label's limitations in ensuring a consistent child viewing experience.

When comparing ad frequency within the same region, we find a substantial decrease in ad exposure through labelling in low policy areas. The mean number of ads per video decreases significantly from 1.7 on unlabelled to 1.1 on labelled content ($p < 0.01$). Surprisingly, the corresponding decrease in high policy regions is less pronounced (1.9 to 1.8 ads per video). Despite the expectation that stronger policy environments should drive more effective use of labelling, ad exposure still remains significant.

| High Policy | | | | Low Policy | | |
| --- | --- | --- | --- | --- | --- | --- |
| | UL | L | | | UL | L |
| inappropriate | 8.3% | 3.6% | | inappropriate | 25.3% | 28.8% |
| child-directed | 6.8% | 19.2% | | child-directed | 6.8% | 8.3% |

**Table 3: Percentage of "inappropriate" and "child-directed" tagged ads across labelled (L) and unlabelled (UL) content.**

**Ad Duration.** When comparing the cumulative unskipped ad durations ("ad time") across labelled and unlabelled videos in both regions, we find that the "made for kids" label has limited impact on the duration of ad content played, while external policy structures have a greater influence. Fig. 5 shows that in less regulated regions, average ad time is consistently higher—184 secs for labelled and 205 secs for unlabelled content—compared to 160 secs and 165 secs in highly regulated regions. The median difference in ad time also follows a similar trend, rising from 0 to 91 secs between labelled and unlabelled content in low policy regions, and from 32 secs to 47 secs in high policy regions. As such, the influence of content labels within a region appears limited while significant variation is observed across different geographies, underlining how the external environment significantly shapes ad choices on YouTube.

Overall, the "made for kids" label's role in limiting ad exposure to children is currently underutilized. Factors such as regional policies and the ad pool composition affect ad frequency and duration, which suggests that the label's efficacy is influenced by the broader advertising ecosystem and regulatory environment.

**Ad themes.** Next, we compare ad patterns thematically between 1500 labelled and unlabelled child-oriented videos, across both regions, to assess the platform's use of labelling in ensuring safer experiences for child audiences [53]. Table 3 shows a comparison of inappropriate and child-directed ads in unlabelled and labelled content across high and low policy regions. In regions with low policy, the label's effectiveness in segregating ad content appropriately is notably weaker. In high policy environments, the labelling practice strongly influences ad choice on child-oriented videos.

In low policy regions, inappropriate ads rise from 25.3% to 28.8% when viewing child content not marked as "made for kids". The parallel increase in high policy areas is from 3.6% to 8.3%, with unlabelled content hosting a significantly greater proportion of inappropriate ads ($p < 0.01$). The same trend holds for child-directed advertising. In strongly regulated areas, the presence of the "made for kids" label increases the prevalence of child-directed ads from 6.8% to 19.2%, whereas child-directed ads remain low in proportion across both content types in low policy regions.

This suggests inconsistent enforcement of the "made for kids" label, warranting further study. Overall, there is a need for more effective utilization of the label to regulate ads shown to children. This involves more robust labelling to cover all child content types, and more consistent advertising on these labelled videos to ensure a uniform viewing experience, across all regions and content styles.

**Risk assessment.** Having identified the high risk of inappropriate exposure on unlabelled videos, particularly in regions with stronger child safety focus, there exists a natural question—how likely is a child to encounter this content? Over 80.6% of our unlabelled dataset has above a million views, and 42.6% of the videos exceed 5 million views. Such popular content often surfaces in search results or recommendations [44]. Importantly, this dataset was constructed using popular child-oriented search terms, reflecting a child's typical usage of YouTube. Simply put, a parent or child who uses such search words to casually pick out a video is likely at a risk of exposure to unlabelled, and thereby, less regulated content.

## 3.3 Google Video Intelligence Analysis

Next, we look to explore current machine learning (ML) solutions that could help mitigate inappropriate exposure for children on YouTube. While the platform claims to use such tools for both video and ad content moderation, our results motivate exploring existing software to understand its effectiveness and application in this context [6]. Given the absence of public disclosure on YouTube's specific ML algorithms, we choose to leverage relevant tools that can provide a reasonable benchmark. We choose the Google Cloud Video Intelligence API, a widely used tool for content moderation, owned by the same parent company as YouTube [33]. This API is used to analyze our manually annotated ads further. For our case, we make use of the "explicit content detection" feature which identifies content generally unsuitable for those under 18 years, including nudity, sexual activities, and pornography [33]. The API annotates video frames, assessing the likelihood of adult content presence using tags like "VERY_LIKELY", "LIKELY", "POSSIBLE", "UNLIKELY", and "VERY_UNLIKELY".

The description of what Google Video Intelligence considers as "explicit content" closely matches a sizeable proportion of the ad sample we tag as inappropriate, particularly those videos annotated with the secondary tags "explicit sexual content", "suggestive content" and "inappropriate clothing". We construct a subset of our annotated ads that can be deemed as "explicit" under this definition, comprising a total of 253 videos. We then analyze this subset using the API to measure the prevalence of explicit content detection. Our analysis reveals that 76.7% of sampled ads contain at least one frame that possibly contains adult content. 26.1% are likely to contain explicit elements, and 11.9% are very likely to do so. The high prevalence of explicit content frames identified by the API in this pool has serious implications—underscoring the grave extent of inappropriateness children are subject to. These frames are considered inappropriate for audiences under 18, making their presence on content for children aged 0 to 8 alarming. This highlights the urgent need for enhanced ad content moderation. Further, these results also confirm the potential of existing ML solutions to help in identifying adult content, as the API is effective in detecting major inappropriate elements. Although YouTube claims usage of ML in content moderation, our findings suggest the need for either more advanced algorithms or a more consistent implementation of existing technological solutions to improve ad video assessment.

## 4 DISCUSSION

Our findings indicate that children face a high risk of getting exposed to unregulated advertising messages on YouTube. To address this issue, we now some discuss recommendations and highlight opportunities for both platform providers and advertisers.

**Tighter platform regulation and proactive ad reporting.** Our results show that 3.6% of ads in high policy regions are inappropriate versus 28.8% in low policy regions, suggesting a strong relationship between lax regulation and the risk of age-inappropriate ads on

YouTube. Studies have shown that violent media consumption impacts children's perception of social norms negatively [16]. There is a need for tighter controls on ad content in child-targeted videos from low policy regions. With 53.8% of "Music" category videos deemed inappropriate, efforts to limit such themes could reduce improper exposure. Once exposure has occurred, the process of reporting an ad on YouTube is cumbersome and involves multiple steps *outside* the video player [2]. We advocate for a proactive, possibly within-player, solicitation of feedback. This enables co-viewing parents to actively contribute to mitigating inappropriate ad exposure for children.

**Need for standardised video content labelling.** A large proportion of YouTube's child content remains unlabeled as "made for kids." YouTube's system, relying on creators to identify their content and using machine learning to find children-targeted videos, seems to miss non-English content, mainly from South Asian children entertainment spaces. This indicates the need for more standardized labelling, with less dependency on self-identification, which creators often avoid due to its restrictive nature. A comprehensive review of child-oriented content is necessary, which may require more manual review, interaction assessments on videos (e.g., comments and viewer demographics), and broader channel evaluations.

**Increased ad relevance.** Our analysis reveals that the child irrelevant content category dominates ad types in our tagging. These ads seem unproductive for (i) *advertisers*, who are not targeting children (or co-viewing parents), (ii) *children*, who get exposed to unnecessary commercial messages, and (iii) *content creators*, who may earn less as children are unlikely to interact meaningfully with such ads [10]. At the same time, child-oriented ads remain low, especially in weaker policy regions, which underscores the need for more such ads as YouTube expands to wider audiences [5]. This offers an opportunity for collaboration between the platform and advertisers to provide age-appropriate content for children.

**Emulating YouTube Kids safeguards.** To promote a more transparent and secure environment, several YouTube Kids safeguards should be replicated on the main app, which has more child viewers [24]. These include the use of distinguishable ad bumpers as animated messages indicating that an ad is about to play or has ended—currently a bleak 7.9% across our entire dataset—as a beneficial first step in ensuring that children are cognisant of the information they are being exposed to. Similarly, disabling within-app access to external advertised sites or showing ad click prompts to alert users about leaving YouTube will reduce the risk of unsupervised exposure to inappropriate or harmful external sites.

**Homogeneity in ad exposure.** Despite child safety policies, high policy areas often have diverse ads due to being consumer hubs. They show more variety to children than less regulated markets, which have a 69% ad repetition due to smaller markets. However, these ads are usually longer. Hence, more consistent policies need to be implemented for child safety, unaffected by factors like external regulation or the ad ecosystem. This can be achieved by limiting the number and duration of ads on child-oriented content relative to the main video length on YouTube.

**Increased metadata on ads.** Currently, ads have varying amount of metadata available. Without a review of the video content itself, it is challenging to gauge the relevance and appropriateness of ads. There should be increased responsibility placed on advertisers to provide metadata such as accurate "made for kids" labelling, relevant description or title, for more standardised assessment of ads. This can enable more responsible advertising and allows for self-reported regulation to help create safer experiences on YouTube.

## 5 RELATED WORK

**Inappropriate video content for children**. A large part of existing research focuses on child-oriented *video content* as opposed to *ad content*. Papadamou et al. [42] developed an ML model that identifies toddler-oriented inappropriate content on YouTube with 82.8% accuracy. Eickhoff and De Vries [26] built a similar classifier that uses video metadata to categorise suitable child content. Related research aims to develop automated mechanisms using video, comments and user-level features for detecting inappropriate videos [12, 42, 43, 47]. Work is predominantly done on YouTube Kids [12, 43, 47], despite its smaller viewer base.

**Inappropriate advertising on child-oriented content**. While previous research has explored advertising within child-oriented YouTube content, a significant gap lies in the lack of studies that analysis regional variations in ad patterns and content labels. Ferreira and Agente [29] uncovered the need for more ad regulation on YouTube Kids, analysing targeted advertising algorithms. Campbell [3] illustrated a range of unfair or deceptive marketing practices on YouTube Kids, in violation of FCC ad regulations. Food and beverage advertising remains a popular focus in works that assess advertisement content on child videos [17, 30]. Liu et al. [36] conducted a large scale study to assess ad patterns and conclude that 26.9% of child-oriented videos contain at least one inappropriate ad. Common Sense Media's work corroborates this frequency, through manual annotation with a holistic code book [20]. Yeo et al. [52] also carried out a small-scale study to highlight the frequency and duration of ads on popular child-directed work, concluding that ad load and age-appropriateness widely varies across YouTube.

Our work aims to address the gap in current literature with a critical *comparative lens* across regions and content labels. We measure the differences between the risk of viewership across "made for kids"' labelled and unlabelled content, to assess YouTube's ad and child protection policy enforcement. Our work involves *both* measurement of ad exposure, and extensive manual annotation to broaden the scope of our analysis, and categorise ads across the lens of appropriate, inappropriate and irrelevant, to answer key questions about the role of ads for child safety on YouTube.

## 6 CONCLUSION

In this paper, we presented the first measurement analysis of how ad exposure and content varies across child oriented videos on YouTube. We analyzed both "made for kids" labeled content and unlabelled content in five strictly regulated and five unregulated countries. Our findings reveal that the safety of a child's YouTube experience is shaped significantly by their external environment and surrounding child safety policies. There also exists a gap in YouTube ad and child protection policy enforcement, indicated by the presence of unlabelled child oriented content with weak ad regulation. Inappropriate ad exposure has serious implications on children, and policy and implementation measures are needed to mitigate this threat.

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

# A APPENDIX

## A.1 Supplementary Details

| Region | Categories (Percentage) | | |
|---|---|---|---|
| | **First** | **Second** | **Third** |
| **High Policy** | People & Blogs (18.4%) | Science & Technology (15.1%) | Entertainment (14.9%) |
| **Low Policy** | Music (36.8%) | People & Blogs (26.7%) | Entertainment (11.6%) |

**Table 4: Top 3 categories of ads.**

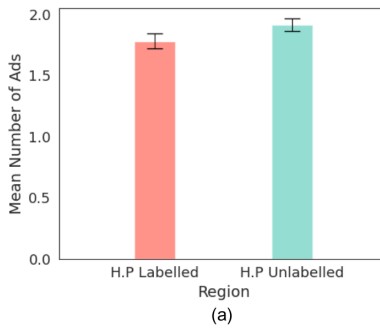 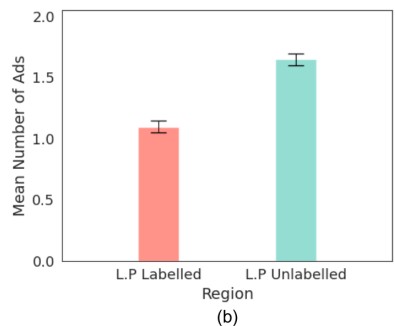

(a)                      (b)

**Figure 6: Mean ad frequency across labelled and unlabelled content in (a) high policy regions and (b) low policy regions. Error bars show 95% confidence intervals.**

## A.2 Criteria and secondary tags for different primary tags

### Table 5: Secondary tags for "Child-directed" category

| Secondary tags | Description |
|---|---|
| Cartoons | Animated shows or movies aimed towards children |
| Storytelling | Videos with story-telling elements (which are not animated). This may include children's shows, vlogs etc. |
| Books and Literature | Videos about reading books, comics, or magazines for children. |
| Family friendly gaming | Videos featuring video games designed for children. |
| DIY & Arts and Crafts | Videos about arts and crafts, including instructions or demonstrations. |
| Toys | Videos that feature promotion, reviews, unboxing, demonstrations, or play sessions with different toys for kids |
| Educational Content | Educational content made for children, e.g. science experiment demos, videos promoting learning apps and games etc. |
| Nursery Rhymes & Music | Rhymes, songs, poems, or musical compositions meant for children |
| Play and Adventure | Videos emphasising physical and adventure activities for children e.g. dancing, rock-climbing, theme parks, camping etc. |
| Health & Hygiene | Health and wellness oriented videos aimed specifically at children e.g. kids' soap, toothpaste, shampoo etc |
| Kid's Fashion | Promoting kid's fashion items such as kids clothing, school bags, shoes and other accessories |
| Movies | Movies for children, including animated films, family-friendly movies, rated G and PG officially |
| Cooking and Food | Videos involving food items or cooking demonstrations targetting children. |

### Table 6: Secondary tags for "Inappropriate" category

| Secondary tags | Description |
|---|---|
| Physical Violence | Videos depicting acts of physical violence including fighting, hitting, physical aggression etc. |
| Interpersonal Violence | Violence or aggression between individuals, such as bullying, harassment, or abusive behaviour, which may or may not be physical |
| Self Harm and Suicide | Videos featuring self-injury, suicide, or any harmful actions inflicted on oneself. |
| Scariness | Content that might be frightening or disturbing for children including horror elements or characters, fire, jump scares, loud sounds or distressing imagery |
| Extreme stunts | Depictions of life endangering or high-risk physical activities and challenges |
| Explicit Sexual Content | Explicit or overt sexual material, activity, or discussions that are not suitable for children |
| Stereotypes | Videos that perpetuate harmful stereotypes, bias, discrimination, or unfair judgments |
| Offensive language | Profanity, hate speech, derogatory or disrespectful language or gestures that are inappropriate for children. |
| Drinking, drugs, smoking | Videos featuring alcohol, drugs, or tobacco products, including vapes |
| Gambling | Videos involving or promoting gambling activities |
| Crude Humour | Content that includes obscene, or inappropriate jokes or humour |
| Display of Weapons | Videos showcasing weapons, firearms, swords or any dangerous objects that can lead to physical harm |
| Inappropriate Clothing | Content featuring excessive or inappropriate skin exposure, wearing revealing or sexually suggestive clothing. Not including clothing worn for activities like running, swimming, wrestling or sunbathing etc. |
| Death | Explicit or graphic depictions of death, corpses, graves etc. |
| Suggestive Content | Videos with romance, or sensual displays of affection, which is not explicit but may still be inappropriate for child viewership |
| Fight Sports | Content related to combat sports or martial arts that may involve blood, or physical harm, which is not suitable for children |

**Table 7: Secondary tags for "Irrelevant" category**

| Secondary tags | Description |
|---|---|
| Home & Office | Videos featuring home or office items, such as furniture, appliances, pet products etc. |
| Health & Wellness | Videos about physical and mental health, nutrition, medication etc. |
| Fashion | Fashion items & accessories like clothing, jewellery, bags etc. |
| Personal Care & Beauty | Videos about beauty and personal care including products such as soap, skincare, makeup etc. |
| Technology & Gadgets | Videos featuring promotion or review of tech devices such as smartphones, laptops, speakers etc. Also includes network services such as mobile and internet |
| Food & Cooking | Videos about cooking and food including recipes, demonstrations, food items etc. |
| Fitness & Exercise | Videos about exercise, diet, weight-loss, athletic gear, equipment etc. |
| Travel & Adventure | Video about travel destinations, adventure activities, as well as elements of tourism, including airlines, trains, hotels |
| Parenting & Family | Videos promoting items for family use or for babies marketed to parents, such as diapers, strollers etc. |
| Education & Learning | Videos about education, learning, academic or professional advice as well higher education institutes and online courses |
| Gaming & Entertainment | Content about video games, TV shows, movies, video streaming or other means of entertainment. |
| Art & Design | Videos about art including painting, crafts, design, art exhibitions etc. |
| Sports & Athletics | Videos about sports, athletes, & sporting events. |
| Music & Performance | Songs, music videos, concerts, including promotions & conversations about music |
| Comedy | Videos featuring comedic skits, stand-up, & humorous content |
| News & Current Affairs | News & discussions about current events |
| Science | Videos about scientific discoveries, advancements, & innovation. |
| Business & Entrepreneurship | Videos about business services, strategies, entrepreneurship etc. |
| Personal Development | Videos offering self-improvement tips, motivation, and life hacks |
| Documentary & History | Documentaries and informative content about historical events |
| Nature & Environment | Videos showcasing nature, wildlife, and environmental conservation |
| Social Issues & Activism | Videos showing activism or raising awareness about social issues. |
| Finance & Legal | Videos about financial management, investments, and legal matters. Includes videos about banks or banking apps. |
| Software | Any software, application or website which provides a service , including e-commerce websites, and management tools |
| Machinery | Ads promoting industrial machinery, tools or hardware |
| Autos & Vehicles | Videos featuring or promoting vehicles or related to automotive topics. Includes online services for buying and selling vehicles. |
| Celebration | Videos referencing communal holidays, ceremonies or festivities |

## A.3 Unlabelled content rubric

**Table 8: Criteria for identifying unlabelled child-oriented content.**

| | |
|---|---|
| **Should include:** | |
| Cartoons | Videos that contain animations specifically created for children. These may include G and PG rated animated movies. |
| Child-centric storytelling | Videos that involve narrating or presenting stories that are not animated, but the main actors are children including child-directed shows, family-friendly movies, vlogs etc. |
| Popular kids' characters | Videos that involve or address popular characters from children's literature, movies or cartoons. |
| Poems & Nursery Rhymes | Rhymes, songs, poems, or music aimed for children/ |
| Family-friendly gaming | Videos featuring video games designed for children, prioritising fun and learning. |
| DIY & Arts and Crafts | Videos about arts and crafts, including instructions or demonstrations, intending and suitable for children |
| Toys | Videos promoting, reviewing or unboxing toys for children. |
| Play and Adventure | Videos about physical and adventure activities for children e.g. dancing, rock-climbing, theme parks, camping etc. |
| External features | Videos with external features that may indicate child viewership for example, comments under videos, video descriptions, or the use of tags that contain children related keywords. Unlabelled videos may also be included from channels for which similar videos have been labelled. |
| **Should not include:** | |
| Violence | Videos featuring violence, physical aggression or harm, weapons, depictions of blood, self-injury or death. These may even include animations or video games that look child-friendly but contain violence |
| Adult animations | Animations that feature sexual themes, nudity, obscene animations meant for adult audiences, or videos describing the design process for creating child animations |
| Adult equipment | Videos that involve the use of hazardous or sharp instruments such as needles, cutters and knives. These may include cooking shows that involve knives or DIY crafts involving scissors or staplers |
| Dark and Scary themes | Content that might be frightening or disturbing for children including horror elements or characters, fire, jump scares, loud sounds or distressing imagery |
| Mature themes | Explicit or overtly sexual material, crude or dark humour, or mature discussions that are not intended nor suitable for children. |
| Offensive language | Sarcastic, derogatory, profane or disrespectful tone, language, gestures or slang. |

