# OpenReview forum: "Analyzing Ad Exposure and Content in Child-Oriented Videos on YouTube"
_ACM.org/TheWebConf/2024/Conference — TheWebConf24 Oral_

### Official Review · Reviewer_APEF · 2023-11-21

**Novelty:** 5
**Technical Quality:** 5

**Review:**

Overall
The paper offers an important look at how the potential exposure of children to ads—particularly those that are not suitable for children—varies across countries with different levels of policy regulation around advertising to children.  The topic and findings are important and may spark valuable discussions about how to reduce cross-country disparities in children’s safety online.

My main critique of the paper is the authors’ repeated potential misattribution of the underlying causes of the results they observe.  They seem convinced that the policy environment explains these differences … when the differences are likely explained by some complex combination of policy environment, economic environment, and algorithmic confounding (how YT’s algos target ads in different countries … which could in turn be influenced by different usage patterns of YT in different countries).  I think this significantly weakens the paper.  I think if the authors simply presented their study as an observational one—where they highlight associations between policy environment and the nature of ads that children might see—and explore a range of reasons that this might be the case (e.g. those above), that would enable the paper to stand on its own as a cautious yet important contribution to the literature.  Without exploring these alternative explanations, though, I worry that the authors are too narrow in their interpretations, and hence, may close doors to future research and possibly inadvertently misinform interventions that should be taken to address the issues they surface.

More details on this critique, with examples and suggestions for changes, can be found below undered “Detailed comments” (look for (--)).

Quality
The study is well-designed and well-executed, but a large quality gap, I think, stems from the critique above.

Clarity
The methods and findings are clearly communicated, but again, the implications/recommendations that truly should be considered are hard to glean because of the authors’ primary focus on policy environment as responsible for the regional differences they find.

Originality
The authors’ focus on a cross-regional/country comparison of children’s potential exposure ot ads on YouTube is novel and constitutes an important contribution to the literature.

Significance
See above — I think this work is commendable, important, and significant, but I worry this significance is hampered by the issues suggested above.

Detailed comments
Introduction
* “going up to 1.8 hours in length” -> there are ads this long??
* (--) “Our study shows that external policy presence drastically affects nature of ads on "made for kids" content” -> the authors can’t be sure that it’s the external policy presence that’s responsible for the difference in nature of ads … there could be many features correlated with country and propensity to implement child ad policies that are responsible (e.g., how ad creators create and distribute ads; etc).  I would soften this to say “external policy presence is strongly associated with the nature of ads …”
	* E.g., differences in ad lengths could be because of different commercial / marketing practices in countries that also happen to have fewer child ad policies in place?  Or perhaps countries with fewer child ad policies are also ones that are less economically developed (as the list of countries suggests), which in turn means fewer ad dollars spent, and perhaps a less targeted ad presentation algorithm … (YT surfaces ads in a more blanket way to increase the chances that someone clicks/engages).  In this case, it’s the country’s economic, not child policy environment, that would be the cause—alongside YouTube’s algorithms.  There are lots of potential confounders here…
	* Another possibility: in places with fewer child ad policies and lower economic development, it’s possible adults and children are more likely to share devices?  If so, could YouTube’s algorithm be confused and unsure of whether or not the viewer is parent or child, based on which videos are being watched from that device/IP address (and hence, less effective in its ads targeting)?
	* Update wording in other places as well: “to determine the role policy presence surrounding child online safety plays in ad exposure” … “These help us in measuring the impact of regulation on ad patterns” … “we find that policy presence does not seem to have a strong influence on ad frequency” etc. … again, hard to know if it’s policy presence or other confounders, like economic development, youtube’s algorithm, general commercial practices, etc.

Methodology
* 2.1 — ”capturing a broad spectrum of content and styles which are likely to attract a large number of young viewers worldwide” …
* 2.1 — how were these keywords selected so that they span relevant topics/areas of interest across the countries in the sample?
* 2 — I am left at the end of this section unclear about how ads are mapped to countries.  Is this something that the ad metadata reveals for each video — which ads were shown, and in which countries?  This seems like a crucial bit of information

Results
* I continue to have a difficult time separate out the potential impacts of policy environment from economic development/capacity on the stated results.  E.g., couldn’t stronger child policy countries, which are also richer, simply have more marketing resources for developed child-directed ads (to explain the 19.2% vs. 8.3% difference)?  I think it’s important that the authors consider these possible alternative explanations instead of pinning results squarely on the policy environment
* 3.2 — “we find that 60.4% of the dataset comprises videos for non-English speaking audiences. Specifically, 33.6% of the videos are in languages indigenous to the subcontinent (Urdu, Tamil, Hindi, Bengali, among others), with 9.3% in European languages (including Spanish, Greek, Polish, Hungarian).” -> possible explanation could be different norms in non English speaking countries around what counts as “made for kids”?  I think it would be helpful for authors to reflect on this …

Discussion and conclusion
* The authors do not reflect on their limitations, which could inform future work on the topic

**Questions:**

Questions are included in the review above.

**Reviewer Confidence:**

3: The reviewer is confident but not certain that the evaluation is correct

**Scope:**

4: The work is relevant to the Web and to the track, and is of broad interest to the community

---

### Official Review · Reviewer_1Dxg · 2023-11-23

**Novelty:** 5
**Technical Quality:** 6

**Review:**

**Pros**
* Age appropriate ads is a relatively understudied phenomenon - well done addressing it!
* The coding process appears thorough - I see no issues that could affect results
* Good results that are well explored, well discussed, and thoroughly analyzed
* The addition of the brief ML study is very good for this paper, good choice

**Cons**
* Statistics are sometimes buried in text, it would be useful to have more figures
* Writing is a bit muddled - it would be helpful to bold main points or otherwise highlight the most important findings

This paper has some (to me) shocking results that are almost hidden behind the data analysis. The set-up seems thorough, albeit quite standard. I’m a little surprised MTurk wasn’t used for more data, but the data coded is sufficient for the analysis. The data analysis itself seems thorough and well analyzed. However, the pure number of statistics and findings in the text without accompanying figures can make it hard to keep track of the important ones. Some minor editing (highlighting the most important numbers, adding figures, etc.) would improve this paper's readability. Further, the results suggest a frequent disregard of its own standards by YouTube. While this is heavily implied by the authors on a few occasions, stating that this raises concerns about what other standards are not being met is the logical next step and I believe the authors are justified in taking this step.

**Quality**: Good. This paper is thoroughly analyzed and methodology seems thorough and well thought out.

**Clarity**: Clarity is hindered somewhat by muddled writing. Using bolded text to highlight important results, more figures, or some other aides would greatly help this.

**Originality**: Mostly original. This area is understudied, but there are a couple of other papers in the same basic vector. Some major results are also shown in this other work, but the results (mainly ad inappropriateness and length) have yet to be addressed by YouTube or (for the most part) other academics or organizations and the authors have decided to again bring them up. This emphasis takes space from their other original results, but is important enough to highlight again.

**Significance**: These results appear highly significant.


Notes:

The last paragraph of page one references a unique unskippable ad of almost 2 hours. If this is not a typo, a little more information here would be useful. If the 15 second limit is broken regularly, then that’s additional context most readers likely don't have (and an interesting tidbit worthy of additional context!) You address this later in the text, but I would clarify this point early on.

I do not follow the second paragraph of page 2. Which subcontinent? Why does this indicate a gap?

In 3.1, it would be useful to take into the account the number of ads when comparing the number of unique ads (are ads repeated more in high or low policy areas?)

I feel like there’s a missed story in here about how YouTube repeatedly breaks its own promises on ad duration, count, and topic. You point this out, but then seem to almost hide it in the background of your statistics. This seems like a big point. If YouTube is breaking these sorts of promises (easy for them to keep but hard for us to verify), what other promises are being broken? You mention this several times, but I think you could make this a stronger point, since it is the logical conclusion of most of your main findings.

There is a little bit of a hidden variable here. Videos in your dataset are popular (world wide, I believe?) but may be more/less popular in a particular region of study. This could potentially influence the ads shown (perhaps those from Morocco do not watch these channels?). Not much you can do here, but worth at least acknowledging.

Figure 5, please use a log scale.

It would be useful to say more about how your study differs from related work. Some of these such as Liu et al. sound similar, and it would be useful if you could spend a few more sentences explaining what you’ve done differently.

I would highlight Table 1 again towards the end of the paper. You have some really fascinating points, but they can get a bit lost in the data. Table 1 is a very clear way to state your findings and referencing it again would remind readers of what you’re trying to say.

**Questions:**

Could you clarify which of your major results you consider completely novel and which you think are drawing more heavily from earlier work?

**Reviewer Confidence:**

3: The reviewer is confident but not certain that the evaluation is correct

**Scope:**

4: The work is relevant to the Web and to the track, and is of broad interest to the community

---

### Official Review · Reviewer_wbTP · 2023-11-24

**Novelty:** 6
**Technical Quality:** 6

**Review:**

In this paper, the authors study the ads that appear on YouTube videos for children. The authors study labeled and unlabeled children's videos---and their associated ads---drawn from high- and low-regulation countries around the world. The authors document a number of problematic and, frankly, embarrassing problems on the YouTube problem.

In general, I like this paper a lot. The topic is critical because kids watch *a lot* of  YouTube. YouTube has reasonable policies about ads targeting kids, but the authors demonstrate that these policies are not being upheld or enforced, especially in non-Western contexts.

The primary issue with this paper is that the authors repeatedly make claims that suggest regulation improves the ads YouTube shows to kids, either because they are shorter or because they are more appropriate. But correlation is not causation. As the authors themselves note, the countries with strong regulations also have stronger economies with more spend on advertising. It may be the case that kids see safer and more relevant ads in regulated markets simply because there are more marketers spending more money to reach children. I am very wary of assertions that regulation is working without strong causal evidence to back up this claim. These claims can be toned down and properly caveated without blunting the overall message of the paper: that YouTube is failing children.

Specific Comments:

There are a lot of tables in the Appendix, but the references to them in the main text are unclear. For example, Section 2.3 refers to a codebook in the Appendix: which table in which section of the Appendix? More precise references would really help.

2.3, Test Coding: The authors state that all authors labeled the ads, but at this stage of blind reviewing the number of authors is unclear. How many labelers were there?

2.3, Final Coding: At this point, I am concerned that the labels were not fluent in the dominant language of many of the assessed countries, such as French, German, and Spanish.

3.1: It's a little weird that Figure 3 is discussed before Figure 2.

3.1, YouTube Ad polices: I don't agree with the framing of some of this analysis. If YouTube says that they have an ad length policy for YTK, then I would only expect that policy to be enforced on YTK. It is not reasonable to expect YouTube to apply to it on main YT---even for videos meant for children. The authors could fix this by changing their framing, i.e., simply stating that YouTube has a different policy on YTK and using it as a benchmark, rather than framing this analysis as a "compliance" assessment.

Somebody uploaded an ad to YouTube that is four hours long? YouTube seems really inept if they aren't catching obviously problematic ads like this.

3.1, Ad themes based on primary tags: Be careful, there is text here that implies that strong child policies may cause more child-relevant ads. However, the authors already identify that this is a problematic correlation, as strong policies are strongly correlated with well-developed and well-funded commercial markets.

The assertion that strong policies prevent or lessen inappropriate ads is also tenuous: YouTube's ad auction will serve whatever ad it deems is most relevant and most lucrative, so in a market where many child-directed ads are available those will be preferentially served. In a market without those ads, the algorithm will serve whatever is available even if it is inappropriate.

3.2, Risk Assessment: This paragraph is repeats information available in the methods section and could be deleted.

**Questions:**

2.1: I'm confused how unlabeled candidate videos were identified. The authors located channels with child-oriented content using keyword search; this part is clear. But then the authors immediately jump to candidate video evaluation. I assume these are videos from the identified channels, but which videos were selected and why from these channels?

2.1: This section could benefit from a lot more detail. How many candidate channels were identified? How many individual candidate videos were labeled? What fraction of these videos were determined to be child-oriented? How many were excluded because they had inappropriate content? What was the inter-rater agreement between the two labelers for whether videos were child-oriented and whether videos contained inappropriate content?

2.3: How did the authors get IP addresses in all 10 target countries (I assume the authors tunneled their crawler through these IPs to scrape geolocal ads)? Did the authors confirm that these IP addresses were being correctly geolocated within the target countries? Did the authors configure Chrome to use an appropriate language in each country (e.g., French in France, German in Germany)?

**Reviewer Confidence:**

4: The reviewer is certain that the evaluation is correct and very familiar with the relevant literature

**Scope:**

4: The work is relevant to the Web and to the track, and is of broad interest to the community

---

### Official Review · Reviewer_BshA · 2023-11-24

**Novelty:** 5
**Technical Quality:** 5

**Review:**

In this paper the authors do an audit of the amount, duration, and content of ads plausibly seen in videos made for children (labelled as such and unlabelled) on YouTube across a variety of countries, from countries with strict(er) polices around social media and children and those with low(er) policies. They find that the YouTube experience (for kids, in their case) is shaped by child safety policies. There is also a gap in the enforcement of child protection policies, and they offer a number of suggestions on how YouTube might take steps to ensure the safety of children on YouTube.

This paper was very well written - it was extremely clear and easy to follow. I really appreciated how the authors described their entire process. One of the biggest strengths of the paper are the importance of the question - we all want to ensure a safe and pleasant viewing experience for our children (whether we have children or not), and that includes ads. Another strength is their comparative research design, analyzing across regions and comparing labelled and unlabelled videos. This is an important audit of YouTube and one that I think should be read.

The biggest question/qualm I have is the suggestion that their analysis focused in on the effect of policy on ads. They compare two groups of countries. One group they label high policy regions (US, UK, France, Germany, Sweden), and the other group low policy environments (Bangladesh, Sri Lanka, Pakistan, Morocco, Venezuela). Looking at those groups, there are a number of other obvious variables that are also likely correlated with ads. Resources, is, of course, the main one, as is institutional capacity, governmental structure, and geopolitical relationships. And the authors briefly mention other differences on page 4 ("One may attribute this to a difference in digital advertising spend, which on average, is reported to be USD 68.9 billion for high policy regions, and only USD 296 million for low policy region.") but these differences require much more discussion, if we are to conclude, as the authors do, that the differences they find are due to policy. It's a tricky causal argument to make.

EDIT: I am keeping my scores the same in response to the authors' comments. I still think this is an interesting and strong paper. I really caution the authors to be careful about the language they use claiming they are doing any sort of evaluation of policy. There are just too many confounders to say anything about policy. In particular, resources. Readers could be easily misled.

**Questions:**

Policy is not that only systematic difference between the two groups of countries you're comparing. How are you dealing with the potential for confounders? This is important because the practical implication of this research is that these countries need stronger policy. But I could come up with different, plausible, stories, which would lead to different policy implications.

**Ethics Review Description:**

No actual human subjects involved - no ethical issues.

**Reviewer Confidence:**

4: The reviewer is certain that the evaluation is correct and very familiar with the relevant literature

**Scope:**

4: The work is relevant to the Web and to the track, and is of broad interest to the community

---

### Official Review · Reviewer_xtpz · 2023-12-01

**Novelty:** 4
**Technical Quality:** 2

**Review:**

This paper analyzes issues in children exposure on either ads or videos on YouTube. They contrast ten countries that adopt different levels and strictness of regulations.

The results show that the existence of regulation reduces the occurrence of inappropriate ads, the compliance to YouTube specifications, and also the duration of ads.

However, there are some issues that indicate that the paper is not mature enough.

The number and characteristics of the countries considered is quite limited. It may have worked better if countries that present varied characteristics were used. When the authors reported some of the phenomena or frauds, my initial impression is that they are associated with the economical scenario, which is not part of the equation.

The annotated dataset of video ads is clearly a contribution. However, I missed a better characterization of this dataset. In particular, I am worried about the size of the ambiguous category, which is not really discussed in the paper.

I believe the analysis has a fundamental caveat by not considering the watching behavior of kids and how these behaviors change as a function of bandwidth availability and connectivity costs. For instance, in some countries there is a significant fraction of pre-paid phones. Further, the percentage of accesses from wifi networks (at home or at work) should be also considered. Other examples of issues are whether popular videos do contain more ads or ads that last longer and the percentage of the video that is effectively watched by users, among others.

In summary, the paper's problem is relevant and challenging, but I believe that it is not cooked enough for acceptance.

**Questions:**

Apparently you chose extremes in terms of regulation, I missed some countries that have some regulation. Did you consider it?

How representative is the ambiguous category? Did you employ any mechanism to mitigate its occurrence?

Why didn't  you consider very popular languages such as Spanish?  It seems to me that comparing English with some very constrained and localized languages is tricky.

Usually you present data for the groups of countries, which are defined by the existence of regulation. What is the variance within each group?

**Reviewer Confidence:**

3: The reviewer is confident but not certain that the evaluation is correct

**Scope:**

3: The work is somewhat relevant to the Web and to the track, and is of narrow interest to a sub-community

---

### Decision · Program_Chairs · 2024-01-22

**Decision:**

Accept (Oral)

**Comment:**

We support AC's recommendation (below) to accept this paper. We encourage the authors to improve the paper for camera-ready based on the feedback from reviewers and AC. We particularly underline the importance of clearly distinguishing between correlation and causation in narrative of the paper for camera-ready and specifically calling out that the results are not causal (something that can be unpacked as part of the Discussion section).

"This paper study the ads that appear on YouTube videos for children. They examine labeled and unlabeled children's videos---and their associated ads---drawn from two group of countries: high- and low-regulation (developed and developing countries). The authors document a number of problematic and, frankly, embarrassing problems that are happening in this platform.

 Strengths:

  - Age appropriate ads is a relatively understudied phenomenon, this is a very important problem
  - The methodology and technical details are solid
  - Interesting results that are well discussed, and thoroughly analyzed

 Weaknesses:
  - Correlation is not causation, the results should be taken with caution (if accepted this must be improved)
  - There are many factors that may explain the results but not all of them are considered (e.g., country economy, the specific countries chosen, etc.)
  - Important details should be highlighted (statistical tests, etc.)

 The problem considered in the paper is very important. If there is no room to accept it, this can also be a very good short paper (4 pages).

 Scope: 4; Novelty: 5; Quality: 5;"